# Chitosan-Coated Liposomes for Intranasal Delivery of Ghrelin: Enhancing Bioavailability to the Central Nervous System

**DOI:** 10.3390/pharmaceutics17111493

**Published:** 2025-11-19

**Authors:** Cecilia T. de Barros, Thais F. R. Alves, Kessi M. M. Crescencio, Jessica Asami, Moema de A. Hausen, Eliana A. de R. Duek, Marco V. Chaud

**Affiliations:** 1Laboratory of Biomaterials and Nanotechnology (LaBNUS), University of Sorocaba, Sorocaba 18078-005, SP, Brazil; thaisfrancinealves1@gmail.com (T.F.R.A.); kessi.moura@uniso.br (K.M.M.C.); 2Biomaterials Laboratory, Faculty of Medical and Health Sciences, Pontifical Catholic University of São Paulo, Sorocaba 18060-030, SP, Brazil; jessicaasami@gmail.com (J.A.); mahausen@pucsp.br (M.d.A.H.); eliduek@pucsp.br (E.A.d.R.D.)

**Keywords:** ghrelin, liposomes, nasal administration, nose-to-brain delivery, bioavailability, central nervous system

## Abstract

**Background/Objectives**: Cachexia is a syndrome characterized by the progressive loss of muscle mass, leading to high morbidity and mortality. Ghrelin (Ghrl) exhibits orexigenic, anabolic, and anti-inflammatory properties with therapeutic potential. However, its low bioavailability limits the efficacy of systemic treatments. This study aimed to develop chitosan-coated liposomes containing Ghrl (CH-Lip + Ghrl) for intranasal administration, allowing quantification of Ghrl brain bioavailability using a system optimized for a labile neuropeptide. **Methods**: The formulation was prepared using thin-film hydration, followed by extrusion and chitosan coating. It was characterized based on morphology, size, zeta potential, stability, encapsulation efficiency, and cell viability. Permeation and mucoadhesion were evaluated ex vivo using porcine nasal mucosa, and cerebral bioavailability was assessed in Wistar rats. **Results**: CH-Lip + Ghrl had an average of 152.4 ± 0.2 nm (evaluated by DLS), a polydispersity index of 0.159 ± 0.018, a zeta potential of +60.8 ± 6.6 mV, and an encapsulation efficiency of 53.2 ± 0.8%, maintaining stability for 180 days. At 1% (*v*/*v*) in culture medium, the formulation retained 73.2 ± 8.4% of the viability in nasal epithelial cells and 81.9 ± 4.8% in neuroblastoma cells. Chitosan coating increased ex vivo mucoadhesion 1.7-fold and permeation 1.3-fold. In vivo, 25 min after intranasal administration, CH-Lip + Ghrl delivered 48.2 ± 8.8% of the dose to the brain, whereas free Ghrl was undetectable. **Conclusions**: The intranasal administration of CH-Lip + Ghrl enhances cerebral bioavailability of Ghrl. This study integrates a chemically labile neuropeptide with chitosan-coated liposomes for direct brain delivery, representing an innovative platform for future translational studies.

## 1. Introduction

Cachexia is a multifactorial syndrome marked by the progressive loss of skeletal muscle mass, sometimes accompanied by a reduction in adipose tissue. This condition is resistant to conventional nutritional support and can lead to functional impairment [1,2]. Cachexia involves a metabolic state characterized by increased protein catabolism and activation of the systemic inflammatory response, both of which contribute to muscle loss and disrupt homeostasis [3]. This syndrome is associated with multiple chronic diseases, particularly cancer types of pancreatic, lung, stomach, and esophageal tumors. It also appears in advanced stages of renal failure, chronic obstructive pulmonary disease, heart failure, chronic liver disease, neurodegenerative disorders, acquired immunodeficiency syndrome (AIDS), tuberculosis, and sepsis [3]. Regardless of the underlying illness, cachexia is a condition associated with increased morbidity and mortality, reduced quality of life, and poorer clinical prognosis [4,5,6]. Despite diverse causes, common pathophysiological features such as systemic inflammation and protein catabolism present potential therapeutic targets [7]. However, current clinical management often focuses on the underlying disease, often leaving cachexia without specific treatments.

Ghrelin (Ghrl), a 28-amino acid orexigenic peptide with a molecular weight of approximately 3.4 kDa, has been investigated as a therapeutic agent for the management of cachexia due to its orexigenic, anabolic, and inflammation-modulating effects. For biological activity, Ghrl requires acylation, with acylated Ghrl representing its only active form. Its primary target is the growth hormone secretagogue receptor (GHS-R1a), which is expressed in regions of the central nervous system (CNS), including the hypothalamus and brainstem. The mechanism of action of Ghrl involves direct stimulation of appetite and modulation of growth hormone (GH) release. Its pharmacodynamic effects include neuroprotection and regulation of energy homeostasis. Modulation of these metabolic and neurological pathways underlies the therapeutic relevance of Ghrl in counteracting catabolic diseases [8,9,10,11]. Nonetheless, its clinical application is limited by pharmacokinetic barriers. Only the acylated isoform of Ghrl is biologically active, but its instability, exacerbated by hydrolysis at the physiological blood pH (7.35–7.45), results in a short plasma half-life and low bioavailability [12,13]. Formulations under investigation utilize intravenous or subcutaneous routes, which are invasive, do not protect against degradation, and compromise patient adherence in long-term treatments [14].

In this context, the nasal route emerges as a promising alternative to conventional administration routes, such as intravenous and subcutaneous injection, which are invasive and rely on systemic absorption followed by crossing the blood–brain barrier. Intranasal administration allows direct transport to the central nervous system via the olfactory and trigeminal pathways, bypassing this barrier. Molecules can reach the brain via olfactory neurons to the olfactory bulb and trigeminal nerves to the brainstem and cortex, enabling direct nose-to-brain transport and supporting the high cerebral bioavailability [15]. In addition, being non-invasive, it enables repeated administration in long-term treatments, constituting a rational strategy for delivering Ghrl to central receptors [16,17].

Although Ghrl exhibits greater stability at the acidic pH of the nasal mucosa (5.5–6.5), local absorption is hindered by enzymatic activity in the mucosa and mucociliary clearance, as well as by diffusion barriers to the central nervous system [18]. To address these challenges, this study developed and evaluated chitosan-coated liposomes containing Ghrl (CH-Lip + Ghrl) as a nose-to-brain delivery system, aiming to enhance permeation, facilitate transport, and improve access to central Ghrl receptors.

## 2. Materials and Methods

### 2.1. Chemicals

Ghrelin (purity ≥ 98%, Cayman Chemical Company, Neratovice, Czech Republic), soy lecithin LS100^®^ (purity ≥ 94%, Lipoid Kosmetik AG, Steinhausen, Switzerland), 1,2-distearoyl-sn-glycero-3-phosphoethanolamine-N-[carboxy-(polyethylene glycol) 2000 (DSPE-PEG2000, Lipoid Kosmetik AG, Switzerland), cholesterol NF (Dishman Group, Ahmedabad, Gujarat, India), ethanol (99%, analytical grade, Anidrol, Diadema, SP, Brazil).

Disodium phosphate (Na_2_HPO_4_, Anidrol, Brazil), monosodium phosphate (NaH_2_PO_4_, Labsynth, Diadema, SP, Brazil), sodium chloride (NaCl, Labsynth, Brazil), phosphate-buffered saline (PBS, pH 5.9 and pH 7.4, isotonic, prepared with ultrapure water from the Milli-Q Plus 185 system, Molsheim, France, with a final conductivity of 18.2 MΩ/cm).

Low molecular weight chitosan (75–85% deacetylated, 50,000–190,000 Da, Sigma-Aldrich, St. Louis, MO, USA), glacial acetic acid (1 N, Dinâmica, Diadema, SP, Brazil).

Eagle’s Minimum Essential Medium (EMEM) with Earle’s Balanced Salt Solution (EBSS) containing non-essential amino acids (Sigma-Aldrich, USA), Dulbecco’s Modified Eagle’s Medium (DMEM), and F12 medium containing non-essential amino acids (Sigma-Aldrich, USA), fetal bovine serum (CultiLab, Campinas, SP, Brazil), aqueous penicillin/streptomycin solution (LGC Biotecnologia, Campinas, SP, Brazil), sodium bicarbonate (Sigma-Aldrich, USA), 0.25% trypsin solution and 1 mM ethylenediaminetetraacetic acid (EDTA, Sigma-Aldrich, USA), trypan blue (Sigma-Aldrich, USA), 3-(4,5-dimethylthiazol-2-yl)-2,5-diphenyltetrazolium bromide (MTT, Sigma-Aldrich, USA), paraformaldehyde solution (8%, Thermo Scientific, Waltham, MA, USA), dimethyl sulfoxide (DMSO, Sigma-Aldrich, USA).

HPLC grade acetonitrile and trifluoroacetic acid (Merck-Millipore, Darmstadt, Germany).

### 2.2. Preparation of Chitosan-Coated Liposomes

#### 2.2.1. Preparation of Liposomes

The liposomes were prepared by phospholipid film hydration, following an adaptation Bangham method [19]. In the organic phase, the lipids were dissolved in absolute ethanol at concentrations of 10 mg/mL soy phosphatidylcholine (LS100), DSPE-PEG2000 (0.6 mg/mL), and cholesterol (0.8 mg/mL), under magnetic stirring at 300 rpm for 10 min. The solution was transferred to a round-bottom flask, and the solvent was removed by rotary evaporation (TE-211, Tecnal, Piracicaba, SP, Brazil) at 65 °C and 350 mmHg for 20 min, creating a dry phospholipid film on the flask walls. The film was hydrated with phosphate-buffered saline (PBS, pH 5.9; 308 mOsm/L) containing Ghrl (70 µg/mL), with gentle agitation to disperse the suspension. This step led to the formation of large multilamellar vesicles (MLVs) as described in the original protocol. Size reduction and transformation into small unilamellar vesicles (SUVs) were accomplished by extruding the suspension through 0.1 µm polycarbonate membranes using a thermostatic extruder (610000-1EA, Avanti Polar Lipids, Alabaster, AL, USA) at 60 ± 5 °C. The suspensions were extruded three times in alternating directions. The final formulation was designated as Lip + Ghrl.

#### 2.2.2. Chitosan Coating

A chitosan dispersion was prepared in 1 N acetic acid and homogenized under magnetic stirring at 300 rpm for 4 h at 25 °C. Three chitosan concentrations (1%, 3%, and 5% *w*/*v*) and three coating times (2, 4, and 8 h) were briefly evaluated to select the optimal conditions. The chitosan dispersion was then added to Lip + Ghrl at a 1:9 (*v*/*v*) ratio, under constant stirring to prevent aggregation and preserve the integrity of the liposome–chitosan complexes. The formulations were maintained under stirring at 300 rpm for the selected coating time to allow complete interaction of chitosan with the vesicle surface.

The absence of free chitosan in the supernatant was confirmed by centrifugation at 25,830× *g* for 30 min at 5 °C (Sorvall ST 16, Thermo Scientific, Waltham, MA, USA), which showed no detectable polymer. The formulation was prepared in PBS (pH 5.9), physiological for the nasal environment, and pH was measured after coating to ensure stability. After coating, the formulations were stored under refrigeration (5–8 °C) until use. The final formulation was designated CH-Lip + Ghrl.

### 2.3. Physicochemical Characterization of Liposomes

#### 2.3.1. Morphology

The morphology of liposomal formulations was assessed by cryogenic transmission electron microscopy (cryo-TEM) on a Talos Arctica microscope (Thermo Scientific, USA) operating at 200 kV and equipped with a 4k × 4k CMOS camera (Walfront, Los Angeles, CA, USA). Samples were applied onto copper grids coated with Lacey carbon film (300 mesh, Ted Pella Inc., Redding, CA, USA), pre-treated to improve droplet adhesion. Vitrification was performed in liquid ethane at approximately −145 °C using the automated Vitrobot Mark IV system (ThermoFisher, Waltham, MA, USA) at 22 °C and 100% relative humidity. The vitrified grids were stored in liquid nitrogen (−196 °C) until analysis and maintained at −173 °C in the microscope chamber during image acquisition. Digital images were captured and analyzed using Digital Micrograph software GMS 3.4 (Gatan Inc., Pleasanton, CA, USA). This process effectively preserves the native liposomal structure, which is crucial for accurate morphological analysis.

#### 2.3.2. Size, Size Distribution, Zeta Potential, and pH

The Lip + Ghrl and CH-Lip + Ghrl formulations were characterized for hydrodynamic diameter and polydispersity index (PDI) by dynamic light scattering (DLS) at a detection angle of 90°. Zeta potential was determined by phase analysis light scattering (PALS) at a 15° angle. Analyses were performed using a ZetaPALS analyzer (NanoBrook 90PlusPALS, Brookhaven Instruments, Holtsville, NY, USA).

Samples were diluted in ultrapure water (1:30, *v*/*v*) and homogenized prior to analysis. Measurements were conducted at 25 °C in triplicate.

Formulation pH was measured using a digital pH meter (TEC-5, Tecnal, Brazil) calibrated with standard pH 4 and 10 solutions at 25 °C. Each sample (2 mL) was homogenized and measured in triplicate, recording the mean ± standard deviation. The electrode was rinsed with ultrapure water (18 MΩ·cm at 25 °C) between readings.

#### 2.3.3. Determination of Encapsulation Efficiency

The encapsulation efficiency (EE) of Ghrl in Lip + Ghrl and CH-Lip + Ghrl was determined using the Bradford colorimetric method [20]. The analytical curve was constructed with Ghrl standards ranging from 3 to 140 µg/mL, and measurements were performed by UV–vis spectrophotometry at 595 nm (RX800, Femto, São Paulo, SP, Brazil).

Formulations were centrifuged at 25,830× *g* for 30 min at 5 °C (Sorvall ST 16, Thermo Scientific, USA). A PBS control solution (pH 5.9) containing Ghrl was subjected to the same centrifugation conditions to verify the absence of free Ghrl sedimentation. The absence of liposomal particles in the supernatant was confirmed by DLS, validating the effectiveness of centrifugation in separating free from liposomal Ghrl.

The sedimented pellet was dissolved directly in Bradford reagent to release the encapsulated Ghrl. Ghrl concentrations in the pellet were determined based on the analytical curve, and encapsulation efficiency was calculated using the following Equation (1):EE (%) = (Amount of encapsulated ghrelin/Total amount of ghrelin) × 100(1)

### 2.4. Stability Studies

The stability of the formulations was evaluated during refrigerated storage (5–8 °C) over 2 years. Hydrodynamic diameter, polydispersity index, zeta potential, and encapsulation efficiency were monitored at the following time points: 0, 7, 15, 30, 60, 90, 180 days, 1 year, and 2 years. Statistical analyses were performed using one-way ANOVA with Dunnett’s post hoc test, comparing each time point to the initial time (day 0), with *p* ≤ 0.05 considered statistically significant, using GraphPad Prism 8.0 software.

### 2.5. In Vitro Cell Viability

#### 2.5.1. Cell Culture

Human cell lines RPMI 2650 (nasal epithelium, BCRJ 0412) and SH-SY5Y (neuroblastoma, BCRJ 0223) were obtained from the Rio de Janeiro Cell Bank (BCRJ, Rio de Janeiro, RJ, Brazil). Both were cultured in an incubator at 37 °C with 5% CO_2_ and controlled humidity.

The RPMI 2650 cell line was maintained in Eagle’s Minimum Essential Medium (EMEM) supplemented with 2 mM L-glutamine, 1% non-essential amino acids, 10% fetal bovine serum (FBS), and 1% antibiotic (penicillin/streptomycin), with medium changes every 2–3 days. The SH-SY5Y cell line was cultured in a 1:1 mixture of Dulbecco’s Modified Eagle’s Medium (DMEM) and Ham’s F-12, supplemented with 2 mM L-glutamine, 1% non-essential amino acids, 1 mM sodium pyruvate, 10% FBS, and 1% antibiotic, with medium changes every 3–4 days.

Subculture was performed when cultures reached 80–90% confluence. Adherent cells were washed with PBS (pH 7.4), dissociated with 0.25% trypsin (with 1 mM EDTA) for 3 min at 37 °C, and enzymatic activity was neutralized with complete culture medium. Cells were centrifuged at 300× *g* for 5 min, resuspended in medium, and counted using a Neubauer chamber. For experiments, RPMI 2650 cells at passage 3 and SH-SY5Y cells at passage 2 were used.

#### 2.5.2. Cell Treatment

Cells were seeded in 96-well plates at densities of 1 × 10^4^ cells/well (RPMI 2650) and 2 × 10^4^ cells/well (SH-SY5Y) in 200 µL of culture medium. After 24 h of adhesion, cells were exposed for 1 h to the experimental conditions described in Table 1, using formulation concentrations ranging from 0.5 to 5% (*v*/*v*) relative to the culture medium.

#### 2.5.3. Cell Viability Assay

Cell viability was assessed using the MTT reduction assay (3-(4,5-dimethylthiazol-2-yl)-2,5-diphenyltetrazolium bromide), a method that measures mitochondrial metabolic activity. After treatment exposure, the medium was removed, and cells were incubated with an MTT solution (0.5 mg/mL) for 1 h (RPMI 2650) or 2 h (SH-SY5Y). The resulting formazan crystals were then solubilized in 200 µL of dimethyl sulfoxide (DMSO), and absorbance was measured at 570 nm using a microplate reader (ELX 800, BioTek Instruments, Winooski, VT, USA).

Formulations were considered viable when cell viability exceeded 70%, as defined in ISO 10993-5: Biological Evaluation of Medical Devices—Part 5: Tests for In Vitro Cytotoxicity [20].

Statistical analyses were performed using one-way ANOVA with Tukey’s post hoc test for group comparisons, with *p* ≤ 0.05 considered statistically significant, using GraphPad Prism 8.0 software.

### 2.6. Ex Vivo Mucoadhesion and Permeation

#### 2.6.1. Preparation of Porcine Nasal Mucosal Membrane

Porcine nasal mucosa was obtained from pig heads supplied by the Angelelli slaughterhouse (Piracicaba, SP, Brazil). Mucosa removal followed the protocol described by Östh et al. [21]. The snout was dissected to expose the ventral nasal concha, and the mucosa was carefully separated from the cartilage using forceps and a scalpel, as this region is most suitable for nasal experiments [22]. Samples were stored frozen in saline solution (0.9% *w*/*v*) and thawed before experiments to preserve tissue integrity.

#### 2.6.2. Mucoadhesion Assessment

The mucoadhesive properties of Lip + Ghrl and CH-Lip + Ghrl were evaluated on porcine nasal mucosa using a TAXTPlus Texture Analyzer (Stable Micro Systems, Godalming, Surrey, UK). The mucosa was secured to the device with plastic tape and nylon thread to ensure stability. Samples (2 mL) were placed on the probe holder and incubated in a water bath at 37 ± 0.5 °C to replicate physiological conditions. A downward force of 0.49 N was applied at 0.2 mm/s, penetrating to 10 mm over 300 s. Following contact, the probe retracted in the opposite direction, and the force required to detach the mucosa from the formulation was recorded, providing measurements of maximum adhesion force, adhesion work, and deformation. All tests were performed in triplicate.

Data were expressed as mean ± standard deviation (*n* = 3). An unpaired, two-tailed Student’s *t*-test evaluated differences between formulations, with *p* < 0.05 considered significant.

#### 2.6.3. Transmucosal Permeation

Permeation of Lip + Ghrl, CH-Lip + Ghrl, and free Ghrl in PBS (pH 5.9) was evaluated using Franz diffusion cells, consisting of donor and receptor compartments separated by porcine nasal mucosa. Formulations were added to the donor compartment (5 mL), while the receptor compartment contained 10 mL of PBS (pH 5.9), maintained at 37 ± 0.5 °C in a thermostatted water bath with continuous stirring.

Samples (1 mL) were collected from the receptor compartment every 2 h over 12 h and replaced with an equal volume buffer to maintain constant volume. As a validation control, PBS (pH 5.9) without drug was added to the donor compartment with mucosa positioned between compartments to exclude potential interference from the mucosa itself.

Permeated Ghrl concentrations were quantified using the Bradford method [23] at 595 nm, based on a pre-established calibration curve. Permeation per area (P/A, µg/cm^2^) was calculated using Equation (2):P/A (µg/cm^2^) = (ghrelin concentration/membrane area) × volume of the receptor compartment(2)

Permeation data were normalized to the total Ghrl dose in the donor compartment (350 µg, defined as 100%) and expressed as percentage permeated for graphical representation.

Data was analyzed by two-way ANOVA to evaluate the main effects of each factor. Multiple comparisons of means were performed to identify statistically significant differences between Lip + Ghrl, CH-Lip + Ghrl, and free Ghrl in PBS (pH 5.9), with significance set at *p* < 0.05, using GraphPad Prism 8.0 software.

### 2.7. In Vivo Brain Biodistribution Study

#### 2.7.1. Animal Care and Handling

The study was conducted with male Wistar rats (Rattus norvegicus), weighing 250–270 g, obtained from a pathogen-free breeding facility (Anilab Paulínia, SP, Brazil; CIAEP no. 01.0264.2014). Animals were maintained under standardized conditions (22 °C, 12 h light/dark cycle) with ad libitum access to food and water.

Six animals were used for the preparation of brain homogenates for linearity studies. For intranasal administration, animals were allocated into two experimental groups receiving either free Ghrl or CH-Lip + Ghrl.

All procedures were approved by the Ethics Committee on Animal Use (CEUA) of the University of Sorocaba (Protocol no. 223/2022) and conducted in accordance with Brazilian Law no. 11.794/2008.

#### 2.7.2. Preparation of Brain Homogenate

Animals designated for brain homogenate preparation were anesthetized with isoflurane (2.5–3%) in an induction chamber, followed by sodium thiopental (0.5 mg/kg, intraperitoneally). Anesthetic depth was assessed by the absence of a nociceptive reflex in response to hind paw pinch. Euthanasia was performed by exsanguination via cardiac puncture, followed by immediate brain collection.

Brain tissues were homogenized in 2 mL of PBS (pH 5.9), centrifuged at 13,585× *g* for 15 min, and the supernatant filtered through a 0.22 µm membrane. The homogenate was spiked with Ghrl at concentrations of 62, 46, 31, 22, 15, 7, and 3 µg/mL to evaluate the linearity of the analytical method.

#### 2.7.3. Ghrelin Determination in Brain Homogenates by HPLC-UV

Ghrl quantification in brain homogenates was performed by high-performance liquid chromatography with ultraviolet detection (HPLC-UV), using a method adapted from Staes et al. [23], selected for its selectivity and suitability for Ghrl quantification.

The mobile phases consisted of ultrapure water (18 MΩ·cm at 25 °C), filtered through a 0.2 µm hydrophilic nylon membrane and degassed in an ultrasonic bath, and HPLC-grade acetonitrile. Chromatographic conditions are summarized in Table 2.

For adaptation to the brain matrix, Ghrl retention times were determined using standard solutions (62.5 and 125 µg/mL) and brain homogenate spiked with Ghrl (62.5 and 125 µg/mL).

The percentage recovery was calculated as the ratio between the experimental concentration (*C*_exp) and the theoretical concentration (*C*_theor), as shown in Equation (3):Recovery (%) = (*C*_exp/*C*_theor) × 100(3)

Method linearity was evaluated using calibration curves prepared in Ghrl standard solutions (62, 31, 15, 7, and 3 µg/mL) and in brain homogenate spiked with Ghrl (62, 46, 31, 22, 15, 7, and 3 µg/mL), allowing comparison of analytical performance in aqueous and biological matrices.

#### 2.7.4. Intranasal Administration for Brain Bioavailability

Animals were anesthetized as previously described, with continuous monitoring of anesthetic depth. While in a standing position, intranasal administration of CH-Lip + Ghrl or free Ghrl in PBS (pH 5.9; 308 mOsm/L) was performed using a micropipette, delivering successive 20 µL aliquots per nostril until a total volume of 1 mL was reached. Intervals between applications were adjusted according to the animals’ spontaneous respiratory cycle to ensure gradual nasal absorption and to minimize swallowing or aspiration of the formulations. Twenty-five minutes after the final administration, animals were euthanized by exsanguination, and brains were collected and processed for HPLC analysis.

## 3. Results and Discussion

### 3.1. Physicochemical Parameters of the Formulations

#### 3.1.1. Morphological Analysis

Cryo-TEM analysis showed that lipid film hydration led to the formation of large multilamellar vesicles (MLVs) with heterogeneous morphology (Figure 1A), consistent with the literature [24]. After extrusion, MLVs were converted into small unilamellar vesicles (SUVs) with homogeneous morphology (Figure 1B), in agreement with Doskocz et al. [25], which demonstrated that extrusion reduces heterogeneity and promotes uniformity.

In the CH-Lip + Ghrl formulation, cryo-TEM micrographs revealed vesicles with continuous membranes, with no signs of fusion or collapse (Figure 2). Two representative micrographs are presented to illustrate the homogeneity of the formulation, demonstrating that the vesicular morphology was consistently observed across different analyzed fields. The technique allows observation of liposomes in a hydrated and vitrified state, preserving their architecture [26]. However, due to the low contrast of PEG and chitosan, it is not possible to directly identify these structures in the bilayer or on the surface [27]. Nonetheless, the analysis complements size determination by showing individual liposomes, with a mean diameter of 141.3 ± 33.1 nm and a coefficient of variation of 23%, below the 30% threshold for liposomes, indicating a narrow distribution and preservation of vesicular architecture. The high zeta potential (+60.8 ± 6.6 mV) results from the extensive protonation of chitosan at pH 5.9 (chitosan pKa ≈ 6.3), which drives dense and stable adsorption of the polymer onto the anionic liposomal surface.

#### 3.1.2. Analysis of Size, Distribution, and Zeta Potential

By DLS analysis, the Lip + Ghrl formulation exhibited a hydrodynamic diameter of 130.25 ± 1.3 nm and a PDI of 0.272 ± 0.027. After coating, the CH-Lip + Ghrl formulation showed an increased diameter of 152.4 ± 0.2 nm, with a PDI of 0.159 ± 0.018, consistent with the presence of the polymeric layer and indicative of efficient coating. Both formulations remained below 200 nm, a range considered favorable for nose-to-brain transport [26]. PDI values below 0.3 further support the homogeneity of the vesicular populations, which is associated with stability and reproducibility [27]. Moslehi et al. [28] reported that uncoated liposomes exhibited a diameter of 141.7 ± 1.7 nm, and after coating, 171.5 ± 0.8 nm, corresponding to an increase of 29.8 ± 1.9 nm. This growth pattern is consistent and comparable to the 22.2 ± 1.3 nm increase observed in our study.

The values obtained by DLS (152.4 ± 0.2 nm) and cryo-TEM (141.3 ± 33.1 nm, coefficient of variation 23%) for CH-Lip + Ghrl are consistent, reflecting the differences in the principles of each technique. DLS measures the hydrodynamic diameter of a particle population in suspension, including the solvation layer, whereas cryo-TEM assesses the physical diameter of individual vesicles in a vitrified field, which may involve a smaller sampling and contribute to the observed variability. Overall agreement between the results demonstrates the reliability and complementarity of both methods in the characterization of nanoliposomes [29].

The chitosan coating altered the surface charge, reversing the zeta potential from −29.9 ± 7.8 mV (Lip + Ghrl) to +60.8 ± 6.6 mV (CH-Lip + Ghrl), indicating polymer adsorption [30,31]. A final chitosan concentration of 0.3% (*w*/*w*) and a coating time of 4 h were selected following an optimization study (Appendix A) aimed at maximizing surface charge while preserving colloidal stability. Although higher concentrations (0.5%) resulted in a higher zeta potential, they also induced an increase in the hydrodynamic diameter (up to 294.31 ± 2.98 nm).

The positive zeta potential also promotes electrostatic interactions between the protonated amino groups of chitosan and the negatively charged sialic acid residues in nasal mucosa mucin [31], enhancing retention and release efficiency [32]. Shirnoush et al. [33] observed that the concentration of chitosan influenced the zeta potential of liposomes, with negative values of −32.5 mV for uncoated liposomes, which became positive (+33.7 mV) after coating. This increase was attributed to the neutralization of the negative charges on the liposomes through interaction with the protonated amino groups of chitosan. Furthermore, Tan et al. [34] reported that chitosan-coated liposomes exhibited a zeta potential of +56.1 ± 4.29 mV, indicating a positive charge due to chitosan adsorption, which is consistent with our results. Similar behavior has been reported by Laye et al. [35], who observed a marked shift in surface charge from −38 mV to +60 mV upon chitosan coating, confirming the electrostatic complexation between the cationic polymer and the negatively charged liposomal membrane.

The formulations exhibited a pH of 5.9 ± 0.2, within the physiological range of the nasal mucosa (5.5–6.5), ensuring tolerability and minimizing irritation [36]. Furthermore, encapsulation in liposomes helps protect ghrelin from degradation during storage, ensuring the reliability of the bioavailability results. Moreover, maintaining the pH within this range preserves the chemical stability of acylated Ghrl, as reported by Staes et al. [23], which is stable under acidic conditions but undergoes accelerated degradation at neutral pH.

#### 3.1.3. Encapsulation Efficiency

The EE of Ghrl was 53.7 ± 0.5% in the Lip + Ghrl formulation and 53.2 ± 0.7% in CH-Lip + Ghrl. The maintenance of EE after coating demonstrates that the process occurred on the vesicle surface without compromising the integrity of the bilayer. This observation is corroborated by studies from Kang et al. [37], who observed that coating with chitosan does not compromise the encapsulation efficiency of bioactives in liposomes.

### 3.2. Formulations Stability

The stability assessment of the Lip + Ghrl formulation demonstrated a progressive destabilization process. The change in zeta potential to −17.2 ± 3.98 mV on day 30 (*p* = 0.0055 relative to day 0) compromised the electrostatic repulsion (Figure 3C). This value is outside the threshold established for colloidal stability of nanosystems, which is greater than +30 mV or less than −30 mV [30]. Consequently, an increase in hydrodynamic diameter (*p* = 0.0291 vs. day 0) and PDI (*p* = 0.0464 vs. day 0) was observed from day 60, indicating vesicle heterogenization and fusion (Figure 3A,B). The structural compromise was corroborated by the continuous decline in encapsulation efficiency, with a sharp decrease after 15 days, reaching an EE of only 20.4 ± 1.5% at 90 days (*p* < 0.0001 vs. day 0) (Figure 3D).

The CH-Lip + Ghrl formulation exhibited superior stability compared to Lip + Ghrl. The chitosan coating acted as a polymeric and electrostatic barrier, delaying vesicle aggregation and fusion. The zeta potential remained above +30 mV for 365 days, with a significant decrease at 730 days (+17.5 ± 11.53 mV; *p* < 0.0001 vs. day 0—Figure 4C). Mean size and PDI remained stable for 365 days, with a significant increase only after 730 days (*p* < 0.0001 vs. day 0; Figure 4A,B). Encapsulation efficiency was maintained up to 180 days, with a significant reduction after 365 days (*p* = 0.0040 vs. day 0) and a marked decrease at 730 days (*p* < 0.0001 vs. day 0—Figure 4D).

The prolonged stability of the CH-Lip + Ghrl formulation aligns with similar systems reported in the literature, supporting the benefit of chitosan coating. For instance, Frigaard et al. [38] reported that chitosan-coated liposomes maintained their stability for up to 420 days. The performance of our formulation, which preserved stable diameter, PDI, and zeta potential above +30 mV for 365 days, corroborates these findings and confirms that chitosan acts as an effective polymeric and electrostatic barrier.

### 3.3. Cell Viability Assessment

The viability of RPMI 2650 (nasal epithelium) and SH-SY5Y (neuroblastoma) cells decreased in a concentration-dependent manner for both cell lines. For CH-Lip and CH-Lip + Ghrl concentrations up to 1%, viability remained above 70% in RPMI 2650 (Figure 5) and above 80% in SH-SY5Y (Figure 6), indicating preserved cellular viability.

In the RPMI 2650 cell line, 2% concentrations reduced viability to 46.93 ± 6.2% (CH-Lip) and 45.19 ± 5.6% (CH-Lip + Ghrl), indicating cytotoxicity. High zeta potential values (>+30 mV) can destabilize cell membranes at elevated concentrations. At concentrations of 3% to 5%, viability decreased to below 35%, demonstrating pronounced cytotoxicity. A similar pattern was observed for the CH-Lip + Ghrl groups at all tested concentrations (Figure 5A). Representative micrographs showed morphological alterations of the cell membrane at higher concentrations (Figure 5B)

In the SH-SY5Y cell line, the viability pattern was similar to that observed in RPMI 2650. Concentrations of 0.5% and 1% maintained viability above 80%, whereas from 2% onward, viability decreased to approximately 51.67 ± 4.2% (CH-Lip) and 52.12 ± 4.3% (CH-Lip + Ghrl). Concentrations of 3% and 5% further reduced viability, indicating evident cytotoxicity (Figure 6A), with corresponding morphological alterations visible in the micrographs (Figure 6B).

The decrease in viability from 2% CH-Lip indicates a critical threshold of 200 µg/mL of phospholipids, consistent with Syama et al. [39], who reported maintenance of viability for neutral liposomes up to 128 µg/mL. Cytotoxicity is attributed to liposomal vehicles and is associated with both phospholipid concentration and liposome surface charge, as also described by Schwendener [40]. High phosphatidylcholine concentrations can alter bilayer fluidity and induce cell lysis.

No significant difference was observed between CH-Lip and CH-Lip + Ghrl (*p* > 0.9999), indicating that Ghrl encapsulation did not affect biocompatibility. Tolerance observed up to 100 µg/mL of phospholipids confirms previously reported patterns [17] and suggests that toxicity at higher concentrations is intrinsic to the vehicle, not the cell line or the encapsulated peptide [15].

### 3.4. Ex Vivo Mucoadhesion and Permeation Data

#### 3.4.1. Mucoadhesion Properties

The CH-Lip + Ghrl formulation exhibited significantly higher adhesion to porcine nasal mucosa compared to Lip + Ghrl. The maximum detachment force increased to 8.283 ± 0.472 g versus 4.823 ± 0.263 g (*p* < 0.0001), and the adhesion work increased to 268.173 ± 14.837 g·s^−1^ versus 192.82 ± 4.839 g·s^−1^ (*p* < 0.0001), with no change in maximum deformation (3.346 ± 1.837 mm vs. 2.872 ± 1.636 mm; *p* = 0.65), indicating that chitosan coating enhances adhesion without altering the formulation’s mechanical properties.

This 1.7-fold increase in adhesion force is consistent with Salade et al. [17], who reported that chitosan-coated liposomes exhibited mucin adhesion of 89 ± 4%, approximately 1.5 times higher than the uncoated formulation (61 ± 4%).

#### 3.4.2. Transmucosal Permeation Profile

As shown in Figure 7, free Ghrl exhibited limited permeation, reaching only 10.9 ± 3.6% over 12 h, highlighting the need for delivery systems capable of overcoming the nasal barrier. Liposomal formulations significantly enhanced permeation (*p* < 0.0001), with Lip + Ghrl reaching 56.3 ± 4.1% and CH-Lip + Ghrl achieving 74.3 ± 2.9% over 12 h.

These results reflect the efficiency of liposomes as a delivery vehicle for Ghrl, whose interaction with biological membranes facilitates mucosal translocation. The superior performance of CH-Lip + Ghrl underscores the contribution of chitosan’s mucoadhesive properties.

The limited permeation of free Ghrl is consistent with the literature data, which describe the low permeability of water-soluble peptides susceptible to enzymatic degradation across the nasal mucosa. Khan et al. [41] reported that intranasal administration of such peptides results in reduced absorption, highlighting the need for delivery systems capable of enhancing permeation.

Liposomal formulations significantly increased permeation (*p* < 0.0001). Liposomes facilitate translocation across the mucosa via membrane fusion or endocytosis. In addition to the liposomal effect, chitosan coating enhances peptide permeation, demonstrating the impact of the mucoadhesive layer in optimizing absorption [42].

### 3.5. Brain Biodistribution Findings

#### 3.5.1. Ghrelin Quantification in Brain Homogenate

Ghrl quantification in brain homogenate was performed by HPLC-UV, using a method adapted from Staes et al. [23], which demonstrated adequate selectivity with a retention time of 33 min (Figure 8).

Linearity was confirmed both in standard solutions (R = 0.9989) and in spiked brain homogenate (R = 0.9977). However, the significant difference in slopes between the standard solution (208.422) and the brain homogenate (28.128; *p* < 0.0001) indicates a matrix effect, as reported in the literature [43], highlighting the need for a matrix-specific calibration curve to compensate for endogenous interferences.

The minimum detectable concentration was 15 µg/mL of Ghrl in brain homogenate, which is high compared to more sensitive methods, such as Liquid Chromatography coupled to Tandem Mass Spectrometry (LC-MS/MS), capable of detecting in the ng/mL range [44].

The recovery percentage was 89.3 ± 8.0% (55.8 ± 5.0 µg/mL experimental vs. 62.5 µg/mL theoretical), with a relative bias of −10.7%, indicating that the method allows quantification of exogenous Ghrl at concentrations above the minimum detectable level. This relative bias is lower than that reported by Staes et al. [23] for Ghrl in standard solution (−29.3%), demonstrating that the method’s performance is consistent with values observed for Ghrl quantification by HPLC-UV.

#### 3.5.2. Brain Bioavailability After Intranasal Administration

Intranasal administration of 70 µg of Ghrl, in 1 mL of CH-Lip + Ghrl formulation or free Ghrl in PBS (pH 5.9; 308 mOsm/L), resulted in brain concentrations of 33.754 ± 6.124 µg/mL for CH-Lip + Ghrl (Figure 9), corresponding to 48.2 ± 8.8% of the administered dose. In contrast, free Ghrl was not detected in the brain homogenate, suggesting either an inability to cross the blood–brain barrier or concentrations below the minimum detectable level.

These findings corroborate the literature indicating the low cerebral bioavailability of free Ghrl and the need for nanostructured systems to enable efficient nose-to-brain transport. Poelman et al. [45] demonstrated that intranasal administration of free Ghrl failed to activate the hypothalamic signaling system, whereas a synthetic analog applied via the same route induced neuronal activation. Han et al. [46] observed that intranasal administration of Ghrl conjugated to gold nanoparticles resulted in a fourfold increase in cerebral peptide levels compared to endogenous expression, reinforcing the potential of nanostructures to enhance central delivery of Ghrl.

The in vivo administration of 1 mL in 20 µL increments allowed the quantification of Ghrl in the brain. The rapid mucociliary transit of 7–15 min limits exposure of the nasal mucosa, while the 24 h in vitro cytotoxicity assay represents a condition at least 96 times more severe. The fraction of 48.2 ± 8.8% of the administered dose detected in the brain further supports the safety of the formulation, as Ghrl did not affect cell viability at any tested concentration. The increased permeation observed for the CH-Lip + Ghrl formulation across ex vivo nasal mucosa, together with evidence from the literature that the olfactory and trigeminal routes constitute the pathways to the brain [47,48], suggests that mucoadhesive liposomes can exploit these routes to traverse the blood–brain barrier and enhance cerebral bioavailability of Ghrl.

## 4. Conclusions

This study provides consistent evidence that chitosan-coated liposomes enhance the bioavailability of Ghrl when administered intranasally. The formulated CH-Lip + Ghrl nanoliposomes exhibited homogeneity and consistent physicochemical stability. Compared to uncoated liposomes, the chitosan coating conferred greater stability and a positive zeta potential, favoring interaction with the nasal mucosa. In ex vivo studies, chitosan increased mucoadhesion and facilitated transmembrane permeation of Ghrl across nasal tissue.

In vivo data show that intranasal administration of CH-Lip + Ghrl results in detectable cerebral bioavailability, an effect not observed with free Ghrl. These findings validate the potential of mucoadhesive, chitosan-coated liposomes to effectively deliver Ghrl to the brain via the nasal route, representing a promising therapeutic strategy for conditions such as cachexia. Future work should focus on optimizing this platform and evaluating its applicability to other neuropeptides and brain-targeted therapies.

While this study demonstrates, for the first time, effective nose-to-brain delivery of ghrelin using CH-Lip + Ghrl, certain limitations should be noted. The in vivo experiments involved a small sample size, and functional outcomes were not assessed, which will be addressed in future studies to further explore the translational potential of this delivery system.

This study includes the enhanced bioavailability of Ghrl via intranasal delivery, increased mucoadhesion and transmembrane permeation by chitosan-coated liposomes, and detectable cerebral delivery, which was not achieved with free Ghrl.

These findings highlight intranasal delivery as a feasible approach to overcome Ghrelin’s limited brain bioavailability, paving the way for future studies in cachexia models.

## Figures and Tables

**Figure 1 pharmaceutics-17-01493-f001:**
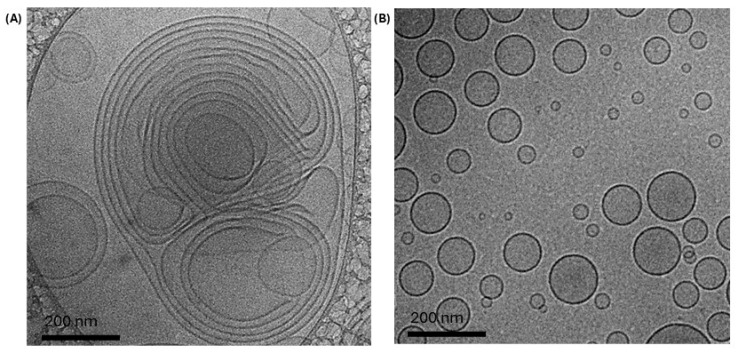
Cryo-transmission electron microscopy (Cryo-TEM) micrographs of liposomes: (**A**) before extrusion, showing large multilamellar vesicles (MLVs), and (**B**) after extrusion, showing small unilamellar vesicles (SUVs).

**Figure 2 pharmaceutics-17-01493-f002:**
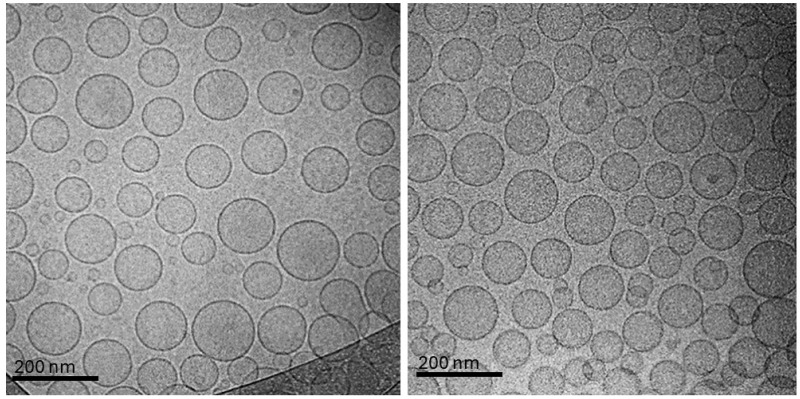
Cryo-TEM micrographs of chitosan-coated liposomes containing Ghrl (CH-Lip + Ghrl).

**Figure 3 pharmaceutics-17-01493-f003:**
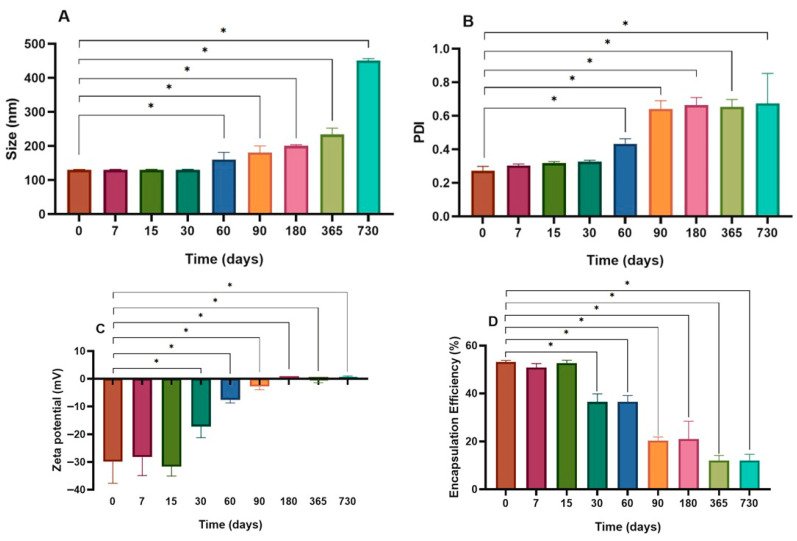
Stability of the Lip + Ghrl formulation during refrigerated storage. Changes in hydrodynamic diameter (**A**), polydispersity index (PDI) (**B**), zeta potential (**C**), and encapsulation efficiency (EE) (**D**) over 730 days. Values are expressed as mean ± SD (*n* = 3). Asterisks indicate statistically significant differences (*p* < 0.05).

**Figure 4 pharmaceutics-17-01493-f004:**
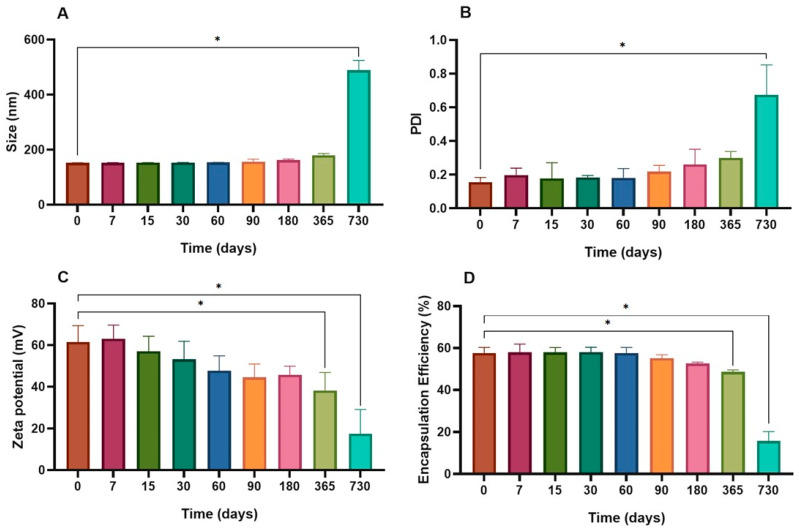
Stability of the CH-Lip + Ghrl formulation during refrigerated storage. Changes in hydrodynamic diameter (**A**), polydispersity index (PDI) (**B**), zeta potential (**C**), and encapsulation efficiency (EE) (**D**) over 730 days. Values are expressed as mean ± SD (*n* = 3). Asterisks indicate statistically significant differences (*p* < 0.05).

**Figure 5 pharmaceutics-17-01493-f005:**
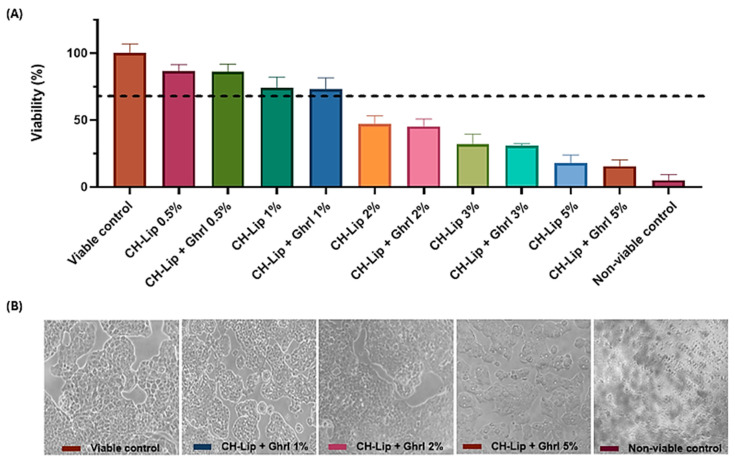
Viability and morphology of RPMI 2650 cells after 1 h of treatment with CH-Lip and CH-Lip + Ghrl (concentration range 0.5–5%). (**A**) Cell viability (%), mean ± SD, shown in the graph; the dashed line indicates 70% viability, adopted as the reference value according to ISO 10993-5 [20]. (**B**) Micrographs of cells exposed to the different tested concentrations and controls.

**Figure 6 pharmaceutics-17-01493-f006:**
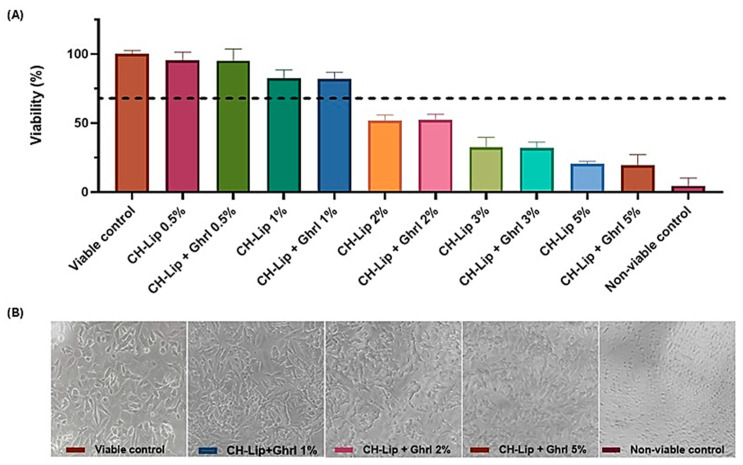
Viability and morphology of SH-SY5Y cells after 1 h of treatment with CH-Lip and CH-Lip + Ghrl (at concentrations ranging from 0.5 to 5%). (**A**) Cell viability (%), mean ± SD, shown in the graph; the dashed line indicates 70% viability, adopted as the reference value according to ISO 10993-5 [20]. (**B**) Micrographs of cells exposed to the different tested concentrations and controls.

**Figure 7 pharmaceutics-17-01493-f007:**
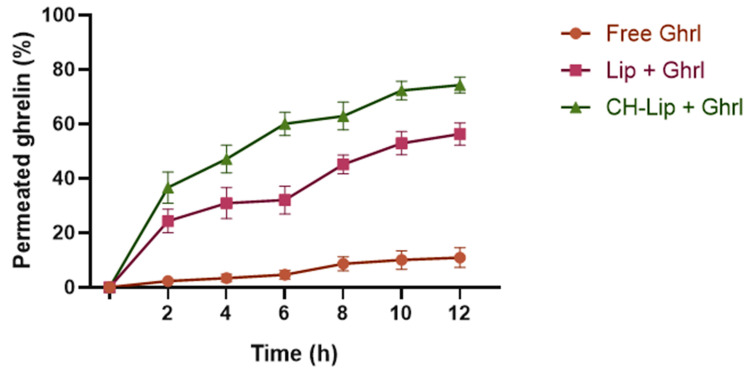
Permeation profile of Ghrl formulations over time in porcine nasal mucosa: free, uncoated liposomes (Lip + Ghrl), and chitosan-coated liposomes (CH-Lip + Ghrl). Values are presented as mean ± SD (*n* = 6).

**Figure 8 pharmaceutics-17-01493-f008:**
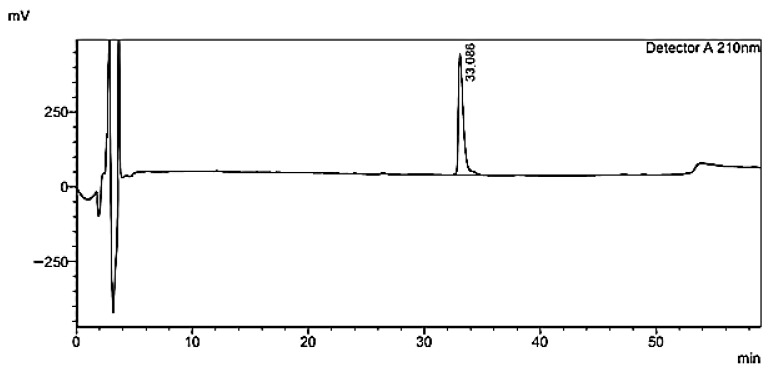
HPLC-UV chromatogram of ghrelin (Ghrl) in a 125 µg/mL standard solution, detected at 210 nm.

**Figure 9 pharmaceutics-17-01493-f009:**
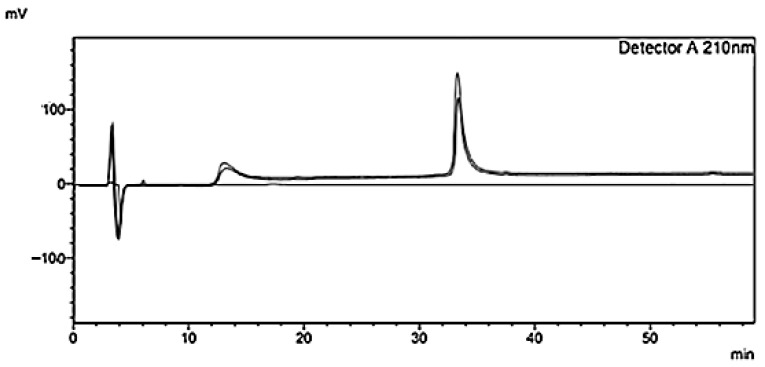
HPLC-UV chromatogram of Ghrl in a brain homogenate sample, showing the peak corresponding to ghrelin detected.

**Table 1 pharmaceutics-17-01493-t001:** Experimental group conditions for the cell viability assay.

Experimental Group	Ghrelin Concentration (μg/mL)	Phosphatidylcholine Concentration (μg/mL)	Paraformaldehyde Concentration (mg/mL)
Viable Control	—	—	—
CH-Lip + Ghrl 0.5%	0.35	50	—
CH-Lip 0.5%	—	50	—
CH-Lip + Ghrl 1%	0.7	100	—
CH-Lip 1%	—	100	—
CH-Lip + Ghrl 2%	1.4	200	—
CH-Lip 2%	—	200	—
CH-Lip + Ghrl 3.5%	2.45	350	—
CH-Lip 3.5%	—	350	—
CH-Lip + Ghrl 5%	3.5	500	—
CH-Lip 5%	—	500	—
Non-viable Control	—	—	40

CH-Lip: chitosan-coated liposomes without ghrelin; CH-Lip + Ghrl: chitosan-coated liposomes containing Ghrl.

**Table 2 pharmaceutics-17-01493-t002:** Chromatographic conditions for ghrelin analysis by HPLC-UV.

Parameters	Conditions
Equipment	Shimadzu Kyoto, Japan HPLC with UV/VIS detector (SPD-M 10A VP), injector SIL-10AD VP
Column	Purospher^®^ (MerckMillipore, Darmstadt, Germany). STAR RP-18 endcapped (250 × 4.6 mm, 5 µm)
Column Temperature	37 °C
Flow Rate	0.95 mL/min
Injection Volume	83 µL
Mobile Phase	A: water + 0.1% trifluoroacetic acid (TFA); B: acetonitrile + 0.1% TFA
Elution Gradient	0 min: 88% A/12% B → 56.14 min: 48% A/52% B

Source: Adapted from Staes et al. [23], using the HPLC method transfer calculator from Sigma-Aldrich.

## Data Availability

The original contributions presented in this study are included in the article/Appendix A. Further inquiries can be directed to the corresponding authors.

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
