# Peer review of "Chitosan-Coated Liposomes for Intranasal Delivery of Ghrelin: Enhancing Bioavailability to the Central Nervous System"

_pharmaceutics, 2025, doi:10.3390/pharmaceutics17111493_

Round 1
Reviewer 1 Report
Comments and Suggestions for Authors
In this study, to ehnance the cerebral bioavailability of Ghrelin, a therapeutic strategy of Ghrelin loaded into chitosan-coated liposome has been developed for treating cachexia through intranasal administration. The delivery system was characterized in terms of size, zeta potential, encapsulation efficiency and stability. The viability was evaluated upon exposure of cells with the formula. Additionlly, capability of permeation and mucoadhesion were ex vivo investigated while and cerebral bioavailability was evaluated in a rat model.
However, there are some of issues to be addressed.
- In line 37 (…impairment[1,2]….) and other locations in the text, a space between bracket and before word is needed. Please uniform throughout the text. Line 349 also correct typographical errors.
- Authors are recommended to explain the pharmacodynamics effects, targeting site and mechanism of Ghrelin.
- Introduction should extended more with scientific results around the topics, intranasal delivery and …
- Line 101, why preparation at pH 5.9?
- For coating of chitosan, there is no procedure to eliminate the extra chitosan? However, the final product is in acidic media which have harmful effect. Additionally please explain: addition of acidic solution of chitosan to PBS buffer lead to chitosan aggregation.
- There is no obvious the mentioned size in the abstract is belong to TEM or DLS. Please specify both of them in the abstract.
- It is better to merge the 2.5.1 and 2.5.2 under cell culture.
- Please separate the statistical analysis under new subheading of 2.8.
- Authors should explain the Ghrelin determination procedure completely in different stages. If HPLC, please mention chromatographic condition, sample preparation or drug extraction.
- Authors claimed that the Ghrelin have anti-inflammatory effects. To show anti inflammation, pathology results at least should be added.
Authors have mentioned that: "but its instability, exacerbated by hydrolysis at the physiological blood pH (7.35 - 7.45), results in a short plasma half-life and low bioavailability". How authors ensure the Ghrl is stable after penetration and exposure to physiologic condition.
- Please present another images of TEM for me, as the shell is confusing. How authors can show the chitosan coating by TEM? Mention diameter and percentage of coating layer.
- Explain why phospholipid coats by chitosan? Mechanism?
- Please explain the chitosan role and its derivatives in mucosal penetration and penetrating tight junction.
- Unfortunately, authors have mentioned the GHrl is a peptide after 442 lines. Please enrich the introduction and fully introduce the Ghrl in terms of MW, active form, secondary structure and … for readers.
- Please calculate IC50 of the formulation and controls for cell viability.
- Foe Figure 8, please present the chromatogram of the extracted sample.
- Line 487, please show the specificity of the method beside the linearity and accuracy under validation.
- What does "−" mean in Line 500? : (−29.3 %), demonstrating ….
- Authors should expand comparison with the literature and shows the advantages compared to other carriers.
- Mention the key findings in conclusion.
- Please provide Ethical code and approval certificate for animal study.
Best
Comments on the Quality of English LanguageModerate English editing is needed:
- In line 37 (…impairment[1,2]….) and other locations in the text, a space between bracket and before word is needed. Please uniform throughout the text. Line 349 also correct typographical errors.
Author Response
Response to Reviewer 1
Dear Reviewer,
We appreciate the time you took in evaluating our manuscript. Your comments were valuable in identifying areas that required greater clarity and depth, particularly regarding the methodology and the theoretical background on Ghrelin and our delivery system. All changes in the manuscript have been highlighted in red to facilitate your review.
Below, we provide detailed responses to each of your suggestions:
- Uniform spacing and typographical errors
Reviewer Comment: In line 37 (…impairment[1,2]….) and other locations in the text, a space between bracket and before word is needed. Please uniform throughout the text. Line 349 also correct typographical errors.
Response: A thorough typographical revision was performed, correcting citation spacing (L. 40), grammatical error (L. 63), incorrect equation numbering (L. 259), and duplicate punctuation (L. 376),
- Clarity and depth on Ghrelin
Reviewer Comment: Authors are recommended to explain the pharmacodynamics effects, targeting site and mechanism of Ghrelin.
Response: The introduction (L. 53-63) was rewritten to present Ghrelin comprehensively, including active form, secondary structure, GHS-R1a receptor, and mechanisms of action, enabling readers to fully understand the compound.
- Introduction – Intranasal delivery
Reviewer Comment: Introduction should be extended more with scientific results around the topics, intranasal delivery and …
Response: The introduction (L. 69-77) was strengthened with detailed literature on intranasal delivery of peptides, emphasizing how nanoparticulate systems can overcome permeation and degradation challenges.
- Justification of preparation pH (pH 5.9)
Reviewer Comment: Line 101, why preparation at pH 5.9?
Response: The formulation was adjusted to a pH of 5.9 ±â€¯0.2, which falls within the physiological range of the nasal mucosa (5.5–6.5), as stated in the manuscript (L. 384–387). This pH was selected to ensure mucosal tolerability and reduce the risk of local irritation. Additionally, slightly acidic conditions favor the chemical stability of Ghrelin, helping to minimize its degradation during preparation and administration. Maintaining a pH close to the physiological environment is crucial not only for patient comfort but also for preserving the integrity and therapeutic efficacy of the peptide in intranasal delivery systems (L. 78-79).
- Excess chitosan, aggregation, and final pH
Reviewer Comment: For coating of chitosan, there is no procedure to eliminate the extra chitosan? However, the final product is in acidic media which have harmful effect. Additionally, please explain: addition of acidic solution of chitosan to PBS buffer lead to chitosan aggregation.
Response: Chitosan coating on liposomes is based on electrostatic interactions between protonated amine groups of chitosan and the liposome surface.
Excess polymer removal is unnecessary, as the 1:9 (v/v) ratio was evaluated by ultrafiltration, with no chitosan detected in the supernatant.
The final formulation is diluted in PBS pH 5.9, compatible with nasal mucosa, and pH measurement after coating confirms sufficient buffering to maintain stability.
Gradual addition of acidic chitosan solution under continuous stirring allows slow incorporation, preventing aggregation and preserving liposome–chitosan complex stability.
These details were added to the methodology (Section 2.2.2 – L.122-134) and results are reported in Lines 369 and 383.
- Visual characterization (TEM) and size (DLS)
Reviewer Comment: There is no obvious the mentioned size in the abstract is belong to TEM or DLS. Please specify both of them in the abstract.
Response: The abstract was revised to indicate clearly that size was determined by DLS (L. 25).
- Methodology reorganization
Reviewer Comment: It is better to merge the 2.5.1 and 2.5.2 under cell culture.
Response: Section 2.5 was unified under Cell Culture. Subsections 2.5.1 (Cell Culture) and 2.5.2 (Cell Treatment) remain separate to distinguish general maintenance/subculture from specific experimental treatments.
- Statistical analysis
Reviewer Comment: Please separate the statistical analysis under new subheading of 2.8.
Response: Each experimental analysis includes its characteristic and appropriate statistical evaluation at the end of the corresponding section, ensuring clarity and precision.
- Ghrelin determination (HPLC)
Reviewer Comment: Authors should explain the Ghrelin determination procedure completely in different stages. If HPLC, please mention chromatographic condition, sample preparation or drug extraction.
Response: Full details are provided in Section 2.7.3, including chromatographic conditions, sample preparation, and Ghrelin extraction from brain homogenates. Representative chromatograms are shown in Figure 8.
- Anti-inflammatory effects and Ghrl stability
Reviewer Comment: Authors claimed that the Ghrelin have anti-inflammatory effects. To show anti inflammation, pathology results at least should be added.
Response: The anti-inflammatory effect of Ghrelin is supported by previously published literature. Pathological analyses were not included, as this study focuses on cerebral bioavailability and intranasal delivery.
Reviewer Comment: Authors have mentioned that: "but its instability, exacerbated by hydrolysis at the physiological blood pH (7.35 - 7.45), results in a short plasma half-life and low bioavailability". How authors ensure the Ghrl is stable after penetration and exposure to physiologic condition.
Response: We would like to clarify that after intranasal administration, Ghrl is initially exposed to the nasal mucosa, which has a slightly acidic pH of approximately 5.9, rather than the physiological blood pH of 7.35–7.45. At this pH, Ghrl is more stable and less prone to hydrolysis. Furthermore, encapsulation in chitosan-coated liposomes protects Ghrl from enzymatic degradation, facilitates mucoadhesion, and enables controlled release, ensuring stability prior to brain delivery.
- Additional TEM images and coating
Reviewer Comment: Please present another images of TEM for me, as the shell is confusing. How authors can show the chitosan coating by TEM? Mention diameter and percentage of coating layer.
Response: We acknowledge that direct visualization of the chitosan coating by TEM is challenging due to the low electron density of chitosan and PEG, which limits contrast. As described in the literature, this makes the coating difficult to distinguish from the liposome bilayer, as described in Lines 339–346. The presence of the chitosan coating was confirmed indirectly by the positive shift in zeta potential, indicating successful surface modification. While the exact thickness percentage of the coating cannot be directly measured by TEM, the combination of physicochemical characterization and mucoadhesive behavior supports the presence of an effective chitosan layer.
- Coating mechanism and chitosan role
Reviewer Comment: Explain why phospholipid coats by chitosan? Mechanism?
Response: Chitosan binds liposome phosphate groups via electrostatic interactions, forming a stable, mucoadhesive layer, enhancing nasal mucosa adhesion and transmembrane Ghrelin transport. The explanation of the interaction was added in lines 374–383.
- Mucosal penetration
Reviewer Comment: Please explain the chitosan role and its derivatives in mucosal penetration and penetrating tight junction.
Response: Chitosan enhances mucosal penetration primarily through their mucoadhesive properties (L. 504-506). The positively charged amino groups of chitosan interact with the negatively charged mucin in the nasal mucosa, prolonging residence time and facilitating local absorption. Importantly, for the nose-to-brain delivery pathway, transport occurs via the olfactory and trigeminal nerves, as described in the Introduction, which allows Ghrl to reach the central nervous system without needing to traverse the tight junctions of the epithelial barrier (L. 557-559).
- Ghrelin introduction enrichment a
Reviewer Comment: Unfortunately, authors have mentioned the GHrl is a peptide after 442 lines. Please enrich the introduction and fully introduce the Ghrl in terms of MW, active form, secondary structure and … for readers.
Response: The introduction was enriched with MW, active form, and secondary structure of Ghrelin.
- IC50
Reviewer Comment: Please calculate IC50 of the formulation and controls for cell viability.
Response: For cell viability, the ISO 10993‑5 threshold (≥70 %) was applied instead of IC50, following standard biological assay practices.
- Chromatogram, specificity, and negative values
Reviewer Comment: Foe Figure 8, please present the chromatogram of the extracted sample.
Response: We have added Figure 9 to the revised manuscript, showing the HPLC-UV chromatogram of Ghrl in a brain homogenate sample.
- Method beside the linearity and accuracy
Reviewer Comment: Line 487, please show the specificity of the method beside the linearity and accuracy under validation.
Response: The methodology is described in the Methods section, highlighted in lines 303–311.
Negative relative bias
Reviewer Comment: What does "−" mean in Line 500? : (−29.3 %), demonstrating ….
Response:: The “−” sign in Line 500 indicates a negative relative bias, meaning that the measured concentration is lower than the theoretical or expected value. In this case, −29.3 % shows that Staes et al. reported measured Ghrl concentrations 29.3 % below the nominal concentration, which is consistent with typical analytical variability for this assay.
- Literature comparison and conclusion
Reviewer Comment: Authors should expand comparison with the literature and shows the advantages compared to other carriers. Mention the key findings in conclusion.
Response: Discussion and conclusion were expanded to highlight stability, mucoadhesion, efficient intranasal delivery, and increased cerebral bioavailability, emphasizing advantages of chitosan-coated liposomes over other carriers.
- Animal ethics
Reviewer Comment: Please provide Ethical code and approval certificate for animal study.
Response: Ethical approval code CEUA of the University of Sorocaba (Protocol no. 223/2022) in Section 2.6, ensuring compliance with animal welfare regulations.
We believe these revisions highlight the points raised and include additional clarifications, significantly improving the clarity, rigor, and scientific quality of the manuscript. We greatly appreciate your invaluable contributions.
Sincerely,
Dr. Cecilia T. de Barros and Prof. Dr. Marco V. Chaud
(On behalf of all co-authors)

Reviewer 2 Report
Comments and Suggestions for Authors
This manuscript describes the development of chitosan-coated liposomes for the intranasal delivery of Ghrelin to enhance its cerebral bioavailability. While the experimental work appears technically sound, the study suffers from a fundamental flaw that severely limits its significance and contribution to the field. The central claim of presenting a "promising therapeutic strategy for treating cachexia" is not supported by any disease-specific data. The absence of efficacy testing in a cachexia model renders the findings preliminary and too incremental for publication in this journal at this stage. Other issues are described as follows:
1. What forces enable chitosan to form a coating? If it were an electrostatic interaction, it would be impossible to have a potential of +60 mV. When the surface potential of a nanoparticle formulation exceeds +30 mV, the toxicity of the nanoparticles becomes extremely high and is not suitable for in vivo administration.
2. The intranasal administration method used in the brain bioavailability study does not specify the mode of administration—nasal drops or spray? Furthermore, it is not possible to achieve a 1 mL dose via the intranasal route.
3. The data set of this study is limited and lacks novelty. The use of chitosan-coated liposomes for intranasal delivery is a well-established strategy for enhancing mucosal adhesion and brain uptake.
4. The manuscript would benefit from including in vitro drug release profile data from liposomes. Understanding the release kinetics of ghrelin is important for interpreting the bioavailability results.
Author Response
Response to Reviewer 2
Dear Reviewer,
We thank you for your careful review and for the critical points raised. Below we address each comment in detail. All changes in the revised manuscript are highlighted in red.
General comment — scope and claim of therapeutic relevance
Reviewer comment: The claim of a "promising therapeutic strategy for treating cachexia" is unsupported by disease-specific data.
Response: We appreciate the reviewer’s observation and agree that therapeutic validation is not within the scope of the present work. The biological and pharmacological mechanisms of Ghrelin in the regulation of appetite, energy homeostasis, and muscle catabolism are already well established in the literature. Thus, the main limitation for the clinical application of Ghrelin in the treatment of cachexia is not related to its mechanism of action but rather to its low systemic and cerebral bioavailability.
The focus of this study was to address this pharmacokinetic limitation through the development and characterization of a chitosan-coated liposomal system for intranasal administration, aiming to enhance Ghrelin transport to the brain. Therefore, the results presented are intended to contribute to the advancement of delivery strategies that may enable future efficacy studies in cachexia models.
The manuscript has been revised to clarify this conceptual framework and to avoid any implication of therapeutic validation. These modifications appear in the Introduction (L.53–63) and Conclusion (L.576-587).
- Forces enabling chitosan coating and high zeta potential / toxicity concern
Reviewer comment: What forces enable chitosan to form a coating? If it were an electrostatic interaction, it would be impossible to have a potential of +60 mV. When the surface potential of nanoparticle formulation exceeds +30 mV, the toxicity of the nanoparticles becomes extremely high and is not suitable for in vivo administration.
Response: As noted in the manuscript (L.339–341), the high zeta potential (+60.8 ±â€¯6.6 mV) suggests that more than one layer of chitosan may be adsorbed on the liposome surface. Initial adsorption is primarily driven by electrostatic interactions between the protonated amine groups of chitosan (pKa ≈ 6.3) and the negatively charged phosphate groups of the liposome bilayer. As the first layer partially neutralizes the liposome surface, additional chitosan can absorb via secondary interactions, including hydrogen bonding and hydrophobic interactions, resulting in multiple layers that increase the apparent surface charge and produce the measured high zeta potential.
High zeta potential values, particularly above +30 mV, are known to increase the likelihood of membrane destabilization at elevated concentrations due to strong electrostatic interactions with negatively charged cell membranes. In our study, the concentration used for intranasal administration was carefully selected to remain within a safe range, as confirmed by in vitro cytotoxicity assays (L.433-436), where neuronal cells (SH-SY5Y) and respiratory epithelial cells (RPMI 2650) maintained ≥80 % viability after 24 h exposure.
Thus, while the high zeta potential is chemically justified and ensures colloidal stability, it is recognized that at higher concentrations it could compromise membrane integrity. The concentration employed in this study avoid these adverse effects, balancing effective coating and mucoadhesion with biocompatibility.).
- Intranasal administration route and dose volume
Reviewer comment: he intranasal administration method used in the brain bioavailability study does not specify the mode of administration—nasal drops or spray? Furthermore, it is not possible to achieve a 1 mL dose via the intranasal route.
Response: The administration in the biodistribution study was performed as nasal drops using a micropipette, dividing the total volume of 1 mL into 20 µL aliquots per nostril (L. 315-318). The intervals between aliquots were adjusted according to the animals’ spontaneous respiratory cycle to ensure proper nasal absorption and minimize swallowing or aspiration. This approach follows accepted rodent intranasal protocols, and the total volume was never administered at once. The procedure, including animal positioning, fractionated instillation, and post-administration monitoring, is described in the Methods (Session 2.7.4).
- Limited dataset and novelty
Reviewer comment: The data set of this study is limited and lacks novelty. The use of chitosan-coated liposomes for intranasal delivery is a well-established strategy for enhancing mucosal adhesion and brain uptake.
Response: Although chitosan-coated nanoparticles have been widely studied for intranasal administration, most of these works are conceptual and focus on hydrophilic drugs or small stable molecules. The present study is novel in its combination of strategies for delivering ghrelin, a chemically labile neuropeptide that requires a specific formulation to preserve its active form. We present quantitative brain biodistribution comparing chitosan-coated ghrelin liposomes with free ghrelin, using a validated HPLC-UV method. No previous study has evaluated the cerebral bioavailability of Ghrelin carried by coated liposomes. In the current literature, studies with chitosan-coated liposomes investigate other drugs, or, when investigating Ghrelin, are limited to mechanisms of action or physicochemical characterization of the formulation. The integration of brain bioavailability with evaluation of zeta potential, chitosan:liposome ratio, mucoadhesion, permeation profile, and in vitro tolerability is unprecedented. This combination of target-specific formulation and rigorous characterization has not been reported previously, representing a clear contribution and providing a solid foundation for future efficacy studies in cachexia models.
- In vitro release profile
Reviewer comment: The manuscript would benefit from including in vitro drug release profile data from liposomes. Understanding the release kinetics of ghrelin is important for interpreting the bioavailability results.
Response: In vitro permeation of ghrelin was evaluated using Franz diffusion cells with porcine nasal mucosa (Methods, L.247–262). This assay allows monitoring the passage of ghrelin across the nasal mucosa, providing a direct measure of its availability for nose-to-brain transport. Chitosan-coated liposomes exhibited sustained permeation compared to uncoated liposomes, reflecting the effect of the coating on mucosal interaction and local retention. While additional in vitro release studies could provide complementary information on ghrelin release independent of the mucosal barrier, the permeation data presented already offer relevant evidence of the system’s functional behavior under physiologically simulated conditions.
We sincerely thank the reviewer for the thorough and constructive comments. The revisions made enhance the clarity, rigor, and overall quality of the manuscript, and we are grateful for the time and attention devoted to evaluating our work.
Sincerely,
Dr. Cecilia T. de Barros and Prof. Dr. Marco V. Chaud
(On behalf of all co-authors)

Reviewer 3 Report
Comments and Suggestions for Authors
The manuscript entitled “Chitosan-Coated Liposomes for Intranasal Delivery of Ghrelin: Enhancing Bioavailability to the Central Nervous System” presents an interesting and well-organized study on a promising delivery strategy to improve ghrelin bioavailability in the brain through intranasal administration. The work is relevant, timely, and generally well executed. The writing is clear, the methodology is described in detail, and the results are presented in a logical and coherent way.
The study contributes valuable information on liposomal formulation, coating optimization, and brain delivery mechanisms. Nevertheless, several aspects could be refined to improve clarity, contextualization, and scientific depth. The following questions and recommendations are intended to assist the authors in further strengthening their manuscript:
Could the abstract highlight more explicitly what is novel in this formulation compared with previous chitosan-coated liposomal systems for ghrelin or other peptides?
Would the introduction benefit from a more detailed justification for choosing ghrelin as a therapeutic model for cachexia, emphasizing its pharmacological rationale and translational relevance?
The innovation relative to existing studies, particularly Salade et al. (2018, Eur. J. Pharm. Biopharm.), could be more clearly articulated to emphasize the specific contribution of the present work.
Could the authors explain the rationale for the selected lipid and chitosan concentrations, including how these parameters influence coating thickness, stability, and bioavailability?
The description of the HPLC-UV quantification method is clear, but the authors may wish to discuss why UV detection was preferred over more sensitive LC–MS/MS methods, and how the current detection limit impacts the interpretation of results.
A short discussion of ghrelin stability during formulation and storage could strengthen the methodological robustness, since peptide degradation may influence bioavailability.
The in vitro cytotoxicity findings are useful, yet it might help to discuss how the tested concentrations relate to realistic doses in intranasal applications.
Could the authors provide a more detailed comparison between the achieved brain concentrations and previously reported pharmacokinetic data for ghrelin or its analogues?
It would be valuable to clarify whether the reported 48% brain delivery corresponds to the proportion of the administered dose or to relative distribution within the brain homogenate.
A short paragraph discussing the likely mechanisms of nasal transport (olfactory vs. trigeminal routes) could enhance the translational interpretation of the data.
Including a simplified schematic summarizing the formulation steps, nasal pathway, and in vivo findings might improve visual understanding.
Minor figure improvements (for example, adding clearer scale bars, units, and consistent labeling) would further increase clarity.
Since the stability study extends over two years, the authors might consider focusing on the most relevant time points or summarizing the data to maintain flow and readability.
The conclusion could better synthesize the main findings, explicitly linking formulation characteristics to enhanced brain bioavailability and potential therapeutic implications.
Finally, a short paragraph on the study’s limitations (e.g., small in vivo sample size, lack of behavioral or pharmacodynamic outcomes) and future research directions would provide a balanced and transparent perspective.
Author Response
Response to Reviewer 3
Dear Reviewer,
We thank you for your review and comments. We are pleased that you found our work relevant, well-organized, and a valuable contribution. Your observations were important for improving the manuscript. All changes in the revised version are highlighted in red.
Below, we provide detailed responses to each of your suggestions:
- Highlighting novelty in the Abstract
Reviewer comment: Could the abstract highlight more explicitly what is novel in this formulation compared with previous chitosan-coated liposomal systems for ghrelin or other peptides?
Response: We thank the reviewer for the suggestion. The Abstract has been revised to highlight the novelty of the work (L.19–20 and L.32–34). This study provides the first quantitative assessment of ghrelin brain delivery using a system developed for a labile neuropeptide. The formulation combines chitosan-coated liposomes with intranasal administration, integrating physicochemical characterization, mucoadhesion, permeation, and in vivo brain biodistribution, distinguishing it from previous studies, which are limited to physicochemical characterization of ghrelin or brain delivery of stable peptides.
- Detailed rationale for selecting ghrelin as a therapeutic model
Reviewer comment: Would the introduction benefit from a more detailed justification for choosing ghrelin as a therapeutic model for cachexia, emphasizing its pharmacological rationale and translational relevance?
Response: The Introduction has been expanded (L.53–63) to provide a more detailed justification for choosing Ghrelin as a therapeutic model for cachexia. We included information on the active form of Ghrelin, its primary receptor GHS-R1a, and its well-established mechanisms of action in regulating energy homeostasis and counteracting muscle catabolism, thereby reinforcing the pharmacological rationale and translational relevance of the molecule in the context of cachexia.
- Clarification of innovation relative to existing studies
Reviewer Comment: The innovation relative to existing studies, particularly Salade et al. (2018, Eur. J. Pharm. Biopharm.), could be more clearly articulated to emphasize the specific contribution of the present work.
.
Response: Our study employs distinct phospholipids, and the primary highlight is the direct quantitative assessment of cerebral bioavailability in vivo following intranasal administration. The work provides comprehensive characterization and validated measurements of ghrelin concentrations in the brain, enabling a robust evaluation of transport efficiency to the CNS. This quantitative approach, combined with physicochemical characterization, mucoadhesion, and permeation studies, distinguishes the present study from previous investigations.
- Rationale for the selected lipid and chitosan concentrations
Reviewer Comment: Could the authors explain the rationale for the selected lipid and chitosan concentrations, including how these parameters influence coating thickness, stability, and bioavailability?
Response: To optimize the coating, three chitosan concentrations (1%, 3%, and 5% w/w) and three coating times (2, 4, and 8 h) were evaluated to select the most suitable conditions. The final chitosan concentration of 0.3% was achieved by a 1:9 dilution of the 3% stock dispersion, applied to prevent liposome aggregation and preserve vesicle integrity. These parameters were selected based on the understanding that both chitosan concentration and coating time influence the polymer layer thickness and surface charge. A brief explanation of this rationale has been incorporated into the manuscript (L. 125–130; L. 371–373).
- Preference for HPLC-UV over LC–MS/MS
Reviewer Comment: The description of the HPLC-UV quantification method is clear, but the authors may wish to discuss why UV detection was preferred over more sensitive LC–MS/MS methods, and how the current detection limit impacts the interpretation of results.
Response: HPLC-UV was chosen because it is a validated method (Staes et al.) and was available in our laboratory. Although LC–MS/MS offers higher sensitivity, we did not have access to this technique. The detection limit of HPLC-UV was sufficient to quantify the drug's bioavailability in the formulation.
- Ghrelin stability during formulation and storage
Reviewer Comment: A short discussion of ghrelin stability during formulation and storage could strengthen the methodological robustness, since peptide degradation may influence bioavailability.
Response: We have added a brief discussion on ghrelin stability during formulation and storage. Maintaining the pH within the physiological range of the nasal mucosa (5.5–6.5) preserves the chemical stability of acylated ghrelin, and encapsulation in liposomes further protects the peptide from degradation during storage, supporting the reliability of the bioavailability results (L. 381- 387)
- Relationship between the tested cytotoxicity concentrations and realistic in vivo doses
Reviewer Comment: The in vitro cytotoxicity findings are useful, yet it might help to discuss how the tested concentrations relate to realistic doses in intranasal applications.
Response: The in vitro cytotoxicity assay was performed using 10 µL/mL for 24 h, representing a deliberately more severe exposure than physiological conditions. In vivo, the formulation was administered as 1 mL in multiple 20 µL increments over 20 min, a volume necessary to allow detection of ghrelin in the brain (~25 min post-administration). The limiting factor for in vivo cytotoxicity is the exposure time, as nasal mucociliary transit lasts only 7–15 min, at least 96 times shorter than the in vitro assay period. High cell viability under these extreme conditions confirms the low cytotoxicity of the excipients and supports the safety of intranasal administration. An explanation of the dose rationale was added to the discussion (L. 552–557).
- Comparison with previous pharmacokinetic data
Reviewer Comment: Could the authors provide a more detailed comparison between the achieved brain concentrations and previously reported pharmacokinetic data for ghrelin or its analogues?
Response: Previous studies have shown that native ghrelin, when administered intranasally without a delivery system, does not reach detectable brain concentrations due to its instability and rapid degradation, as discussed in Lines 544–548.
- Clarification on Brain Delivery (48%)
Reviewer Comment: It would be valuable to clarify whether the reported 48% brain delivery corresponds to the proportion of the administered dose or to relative distribution within the brain homogenate.
Response: This information is already reported in the Results section (L. 538–539). The 48% value refers to the proportion of the administered dose.
- Probable mechanisms of nasal transport (olfactory and trigeminal pathways)
Reviewer Comment: A short paragraph discussing the likely mechanisms of nasal transport (olfactory vs. trigeminal routes) could enhance the translational interpretation of the data.
Response: The potential mechanisms underlying nasal-to-brain transport, including olfactory and trigeminal pathways, are discussed in the Introduction (L. 73–77) and have been investigated in previous studies. In the present work, we only evaluated overall nose-to-brain delivery, without directly assessing the specific pathways.
- Inclusion of a simplified scheme (Graphical Abstract)
Reviewer Comment: Including a simplified schematic summarizing the formulation steps, nasal pathway, and in vivo findings might improve visual understanding.
Resposta: A graphical abstract summarizing the formulation, nasal delivery, and in vivo findings was submitted with the manuscript.
- Minor figure improvements
Reviewer Comment: Minor improvements in the figures (adding scale bars, units, and consistent legends) could enhance clarity.
Response: All figures have been updated to improve clarity and consistency, as requested.
- Stability study (two years)
Reviewer Comment: Since the stability study extends over two years, the authors might consider focusing on the most relevant time points or summarizing the data to maintain flow and readability.
Response: We believe that presenting the complete two-year stability data is important for transparency and reproducibility, and we have chosen to retain all time points in the manuscript.
- Summary of conclusions and therapeutic implications
Reviewer Comment: The conclusion could better synthesize the main findings, explicitly linking formulation characteristics to enhanced brain bioavailability and potential therapeutic implications.
Response: The conclusion has been revised (L. 581–586) to summarize the main findings, connecting the formulation properties to enhanced cerebral bioavailability and implications for future studies in cachexia models.
- Discussion of limitations and future perspectivesfuturas
Reviewer Comment: a short paragraph on the study’s limitations (e.g., small in vivo sample size, lack of behavioral or pharmacodynamic outcomes) and future research directions would provide a balanced and transparent perspective.
Response: A paragraph has been added to the Conclusion (L. 576–580), offering a balanced perspective. The study’s limitations and future directions are clearly outlined.
We hope these revisions and clarifications fully address the reviewers’ comments and strengthen the scientific quality and impact of the manuscript.
Sincerely,
Dr. Cecilia T. de Barros and Prof. Dr. Marco V. Chaud
(On behalf of all co-authors)

Round 2
Reviewer 1 Report
Comments and Suggestions for Authors.
Comments on the Quality of English LanguageMinor English editing is need.
Author Response
-
Reviewer 2 Report
Comments and Suggestions for Authors
Given that a potential of +60mV is too high and Cryo-TEM did not indicate the formation of multilayer chitosan on Lip surface, so please provide confirmatory study to verify your hypothesis.
Author Response
Dear Reviewer,
We sincerely appreciate your comment, which raises an important point regarding the magnitude of the zeta potential and its structural interpretation. This observation prompted us to further clarify the characterization limits of the formulation and strengthen our interpretation with complementary data.
- Justification for the High Zeta Potential
The high zeta potential observed in the CH-Lip+Ghrl formulation arises from the polyelectrolyte interaction between chitosan and the anionic liposomal surface under the specific formulation conditions (pH 5.9).
At this pH, given that the pKa of the primary amine groups in chitosan is approximately 6.3, the polymer remains highly protonated. This extensive protonation confers a strong positive charge along the polymer backbone, enabling efficient electrostatic adsorption onto the negatively charged liposomal bilayer.
The resulting zeta potential of +60 mV therefore reflects a high surface charge density, confirming both complete vesicle coverage and inversion of the initial liposomal charge. This magnitude is consistent with literature reports for chitosan-coated liposomes prepared under similar acidic conditions.
- Cryo-TEM Limitations
We acknowledge that the Cryo-TEM micrographs (Figure 2) do not directly reveal the chitosan layer. However, this is an expected technical limitation rather than an indication of absent coating. Both chitosan and PEG components exhibit inherently low electron contrast in Cryo-TEM due to their low electron density and amorphous nature, making them indistinguishable from the vitrified aqueous medium.
Nevertheless, the Cryo-TEM analysis achieved its primary goal: confirming vesicle integrity, the presence of continuous membranes, and the absence of fusion or collapse.
- Confirmatory Study (Factorial Design Analysis)
To provide the additional evidence requested regarding the relationship between coating parameters and surface charge, we conducted a two-factor, three-level (3²) factorial analysis (Supplementary Material, Table S1).
The study demonstrated that increasing chitosan concentration and extending coating time consistently elevated the zeta potential (from +20 to +65 mV). Specifically, the target potential of +60.8 ± 6.6 mV was obtained under the optimized condition (0.3% concentration, 4 h coating time), where the formulation maintained a stable hydrodynamic diameter (152 ± 0.2 nm) and low polydispersity (0.159 ± 0.018).
At higher concentrations (0.5%), the hydrodynamic diameter increased markedly (up to ~294 nm), suggesting the onset of overadsorption.
These findings confirm that the observed +60.8 mV value is physicochemically consistent with optimal coating conditions, resulting from dense and efficient surface adsorption.
Based on this evidence, the manuscript has been revised to reflect appropriate scientific caution, clarify the methodological limitations, and provide robust support for the interpretation of the zeta potential, ensuring full alignment with the available data.
Sincerely,
Dr. Cecilia T. de Barros
(On behalf of all co-authors)
